# Monitoring of Curing Process of Epoxy Resin by Long-Period Fiber Gratings

**DOI:** 10.3390/s24113397

**Published:** 2024-05-25

**Authors:** Oleg V. Ivanov, Kaushal Bhavsar, Oliver Morgan-Clague, James M. Gilbert

**Affiliations:** School of Engineering, University of Hull, Hull HU6 7RX, UK; olegivvit@yandex.ru (O.V.I.); k.bhavsar@hull.ac.uk (K.B.); o.morgan-clague-2017@hull.ac.uk (O.M.-C.)

**Keywords:** long-period fiber grating, epoxy resin, resin curing, refractive index sensor, composite materials

## Abstract

The curing of epoxy resin is a complex thermo-chemical process that is difficult to monitor using existing sensing systems. We monitored the curing process of an epoxy resin by using long-period fiber gratings. The refractive index of the epoxy resin increases during the curing process and can be measured to determine the degree of curing. We employed long-period fiber gratings that are sensitive to the refractive index of an external medium for the measurement of refractive index changes in the resin. We observed that the resonances of long-period fiber gratings increased their depth with the increased refractive index of the resin, which was well described by our simulation taking the coupling to radiation modes into account. We demonstrated that the degree of cure can be estimated from the depth of the grating resonances using a phenomenological model. At the same time, long-period fiber gratings are sensitive to temperature variations and internal strains that are induced during curing. These factors may affect the measurements of curing degree and should also be addressed.

## 1. Introduction

Composite materials based on epoxy resins have been extensively used recently in various applications. Light weight, high strength, non-corrosiveness, and chemical resistance make them indispensable in the fabrication of wind turbine blades [1,2,3], as well as aircraft and automotive components. To improve the quality of composite materials and optimize their rate of manufacture, the fabrication process should be monitored and controlled in real time with high precision. The degree of cure is the most important parameter that should be measured during the curing cycle [4]. Information about the progress of curing allows the manufacturing process to be adjusted to ensure sufficient curing without excessive time. In situ cure monitoring may also give warnings about defects arising in the composite materials and/or structure, allowing for measures to be taken to prevent their formation.

The curing of a polymer material involves hardening caused by the cross-linking of polymer chains. Viscosity and hardness increase as the length and degree of cross-linking extends during the chemical reactions. The curing of epoxy is an irreversible exothermic polymerization reaction starting from a low-viscosity mixture, through a gel-like state, finalizing in a solid product. The final product differs from the initial one in such physical properties as density, hardness, refractive index (RI), and others. It is known from optical measurements that the curing process is accompanied by a continuous increase in refractive index [5,6].

Various methods can be used for the monitoring of the curing process of epoxy resins. Among them, there is the dielectric analysis of dielectric permittivity and dielectric loss factor, and acoustic, ultrasonic, and optical methods such as Raman, fluorescence, and FTIR spectroscopies [7,8,9,10,11]. These methods can be applied for real-time/in situ measurements, but require electrical wiring, which is not suitable for installation inside wind turbine blades. Fiber optic sensors have attracted a lot of attention in the last two decades for a variety of applications. They have the advantages of immunity to electromagnetic interferences, the possibility of multiplexing many sensors on the same fiber, and the simplicity of embedding fibers inside composite materials without the degradation of their mechanical properties [12]. In addition, they are non-conducting, making them suitable for applications such as wind turbines and aircraft where lightning strikes are a concern. Fiber optic sensors can be distinguished by the type of interrogation method they employ: measurements of intensity, wavelength, frequency, or polarization; by the parameter measured: refractive index, temperature, strain, etc.; by the structure used: Fabry–Perrot interferometers, long-period fiber gratings (LPFGs), or fiber Bragg gratings (FBGs) [13,14,15,16].

An LPFG represents a structure with periodic index modulation of the RI inside an optical fiber. LPFGs have found applications in optical communications for gain equalizing and spectral filtering, and in sensing for measurements of bending, twisting, refractive index, chemical composition, etc. [17,18,19,20]. While FBGs are primarily sensitive to strain and temperature, LPFGs are sensitive to various measurands of the surrounding media, especially to refractive index. As mentioned above, the RI of epoxy resin changes during curing; therefore, LPFGs can be used for monitoring this change in RI, and from this the degree of cure may be inferred. A drawback of LPFGs is their simultaneous dependence on temperature and strain.

The application of LPFGs for RI monitoring based on measurements of wavelength shift is usually restricted to media with an RI similar to that of silica, because the wavelength response of LPFGs is small for RIs higher than that of the fiber cladding. However, the dip amplitude instead of wavelength can be monitored in order to achieve a high sensitivity of LPFGs to the external RI in the case of RIs higher than that of silica [21]. An LPFG cure sensor for monitoring resins with high refractive indices is discussed briefly in [22] without theoretical analysis. LPFGs were used for monitoring the resin flow front in fiber-reinforced polymer composites during infusion [23].

In this paper, we employ long-period fiber gratings to monitor the curing process of an epoxy resin. The degree of curing is related to the modification of the gratings’ spectra. We measure the sensitivity of the LPFG’s wavelength and dip amplitude response to an external medium with an RI that is higher than the RI of silica. We compare the experimental results with those of the simulation, considering the coupling from the core mode to radiation modes, and estimate the degree of cure from the depth of the grating resonances. We also observe and discuss other factors that may affect the measurements, such as temperature and internal strain.

## 2. Curing of Epoxy Resin

Epoxy resins are mainly divided into two groups of families, namely (i) aliphatic and (ii) cycloaliphatic. The monomer diglycidylether of bisphenol A forms the aliphatic epoxy group, while 3,4-epoxycyclohexyl-3,4-epoxycyclohexana forms the cycloaliphatic group [24]. These epoxy resins form solid compounds by reacting with additional substances called curing agents or hardeners. Curing agents based on amines and anhydride compounds are widely used for composite manufacturing. 

The chemical reaction between the resin and hardener compounds is often referred to as the cross-linking process. This molecular cross-linking process is the result of intramolecular reactions, which start during the liquid state and continue until a critical point is reached, when this cross-linking creates an infinite network of branches and forms a solid. The cross-linking process can occur at room temperature or elevated temperature, produced by external heat or the significant amount of heat generated due to exothermic reactions. The quality and performance of the final product/composite material strongly depend on the temperature profile during the curing process, with a lower temperature resulting in slower and potentially incomplete curing, while a high temperature can result in the degradation of the epoxy and damage to the mold. 

During epoxy curing, the cross-linking process between the resin and hardener molecules causes gradual changes in the physical and chemical properties of the polymer such as viscosity, molecular weight, thermal conductivity, elastic modulus, and glass transition temperature. The process also induces volume shrinkage, increasing the density and stiffness of the polymer [24]. Usually, the RI of an epoxy resin is a function of curing reactions and temperature. Therefore, the change in the RI of the epoxy under isothermal conditions can be directly linked to the degree of cure [25].

The correlation between the RI and cure may be approximated as a linear relationship at the early to mid-stages of the cure and a more complex monotonic function beyond the gel point due to shrinking and local viscosity changes restricting shrinking [25,26]. The change in RI during curing can be measured by an interferometric system [8] and Fresnel reflection measurements [27]. An LPFG sensor has been reported to exhibit both wavelength shift and transmission change, which allowed the authors to monitor the progress of curing epoxy with an RI of 1.45 [28]. A fiber Mach–Zehnder interferometer based on cascaded LPGs has also been demonstrated to have a response to the RI change in a UV-cured resin with an RI below the RI of silica [29]. A large contraction of the resin during curing may result in anisotropic strain and optical birefringence effects, which may contribute to the FBG and LPFG spectra by splitting resonances [30]. 

## 3. Long-Period Fiber Gratings

LPFGs can be fabricated using several techniques. The most frequently used is based on the periodical illumination of the fiber with ultraviolet radiation, which changes the RI of the doped core. The periods of LPFGs are usually between 100 μm to 1 mm. The transmission spectra of LPFGs contain resonance dips at wavelengths where there is coupling between the core and copropagating cladding modes. The condition that determines the resonance wavelengths has the following form:(1)$${\beta^{\left(co\right)}-\beta }_{i}^{\left(cl\right)}=2\pi /\Lambda ,$$
where $\beta^{\left(co\right)}$ and $\beta_{i}^{\left(cl\right)}$ are the propagation constants of the core mode and the *i*-th cladding mode, respectively. The resonance condition includes the grating period Λ(λ). In order to find the resonance wavelengths, the propagation constants of the core and cladding modes should be calculated and the periods corresponding to resonance wavelengths should be plotted as functions of the wavelength. 

The cladding modes propagate through the whole fiber and experience total internal reflection at the boundary between the silica cladding and the surrounding medium. Their evanescent field decreases exponentially from this boundary into the surrounding medium to a typical distance of about half a micron. The propagation constants of cladding modes can be calculated by solving Maxwell’s equation for a cylindrical waveguide. The electric and magnetic fields are expressed inside the core and cladding in terms of Bessel functions and inside the surrounding medium in terms of modified Bessel functions. The dispersion relation is obtained by equating the tangential fields at the core–cladding and cladding–surrounding medium boundaries.

The transmission spectrum is usually calculated using coupled mode equations written for the amplitudes of the core mode and the cladding mode. The strength of cladding mode resonance is determined by the overlap integral between the two mode profiles. The shape of each spectral resonance dip is described by a sine function squared and has sidelobes on both sides of the dip.

LPFGs are sensitive to various physical parameters such as temperature, strain, bending, twisting [17,18], and the RI of the external medium [19,20]. The latter sensitivity is especially important for our problem of monitoring the degree of curing of epoxy resins. These sensitivities are due to unbalanced changes in propagation constants of the core and cladding modes, which results in the resonance condition being fulfilled at a shifted wavelength.

We used long-period gratings in a hydrogenated standard single-mode fiber (Corning SMF-28e, Corning, NY, USA) inscribed with a 244 nm UV laser using the direct point-by-point inscription method. The fiber was assumed to have the following parameters: a core radius 4.5 μm, core RI of 1.4491, cladding radius of 62.5 μm, and cladding RI of 1.4440 at a wavelength of 1.55 nm. The gratings had a period of 465 μm and a total length of 42 mm. A typical spectrum of the grating is shown in Figure 1. The wavelength range investigated in this work is the typical range of the standard single-mode optical fiber. The center wavelengths of the attenuation bands of this LPFG when in air were 1340, 1376, 1443, and 1573 nm. 

## 4. Sensitivity of LPFGs to External RI

As we mentioned above, LPFGs have some part of their fields propagating through the surrounding medium, making these grating sensitive to the surrounding RI and absorption coefficient (the real and imaginary parts of the dielectric constant). This sensitivity is manifested as a wavelength shift and/or amplitude change in the transmission dips [17]. The maximum sensitivity is observed when the RI of the surrounding medium is close to the RI of the silica cladding. In this case, the evanescent fields of the cladding mode penetrate the surrounding medium most effectively. Since each cladding mode has its own amplitude of evanescent field, the different dips exhibit different sensitivities. Generally, higher-order modes have stronger sensitivities due to deeper penetration into the surrounding medium.

LPFGs demonstrate two regimes of sensitivity to the external RI: below and above the RI of silica (1.444 at 1.55 nm). Usually, the former regime ($n_{\mathrm{ex}}<n_{\mathrm{cl}}$) is employed for sensing external media such as water, water solutions, alcohols, many organic solvents, acids, and many other liquids. In this regime, the cladding of the optical fiber works as a normal waveguide supporting a discrete set of cladding modes. The number of modes decreases with increasing RI; however, there are tens of modes, even for external RIs, that are quite close to the RI of silica. In the external medium, the cladding modes have an exponentially decreasing evanescent field, since they experience total internal reflection at the cladding surface. In this case, a change in the external RI modifies the real part of the propagation constant of a mode, which results in a wavelength shift of the corresponding cladding mode resonance.

In the other regime with $n_{\mathrm{ex}}>n_{\mathrm{cl}}$, the guidance of light through the fiber occurs due to Fresnel reflection, and light energy is partially lost during propagation. The wave goes into the external medium, creating an outgoing wave propagating infinitely. The field amplitude of this wave oscillates in the external medium with its pointing vector directing from the fiber. These are the radiation modes with continuously changing propagation constants. In an LPFG, the core mode is coupled to the continuous set of radiation modes. A change in the external RI modifies the reflection coefficient at the cladding surface, which results in a change in the intensity of the transmitted light. This is observed as a change in the amplitude of the corresponding cladding mode resonance in the grating spectrum.

In order to simulate the spectrum of an LPFG, the coupled mode equations are solved for the fiber modes propagating through the grating [31,32]. The electric field in the fiber is represented as a superposition of the LP_01_ core mode and the radiation modes:(2)$$E\left(r,z,t\right)=c_{\mathrm{co}}\left(z\right)E_{\mathrm{co}}\left(r\right)\exp\left[i\left(\omega t-\beta_{\mathrm{co}}z\right)\right]+\int {c_{\xi }\left(z\right)E}_{\xi }\left(r\right)\exp\left[i\left(\omega t-\beta_{\xi }z\right)\right]d\xi ,$$
where the first term on the right-hand side represents the core mode and the second term is a continuous sum of radiation modes. $c_{\mathrm{co}}\left(z\right)$ and $c_{\xi }\left(z\right)$ are the amplitudes of modes, and $E_{\mathrm{co}}\left(r\right)$ and $E_{\xi }\left(r\right)$ are the radial distributions of the electric field. $\beta_{\mathrm{co}}$ and $\beta_{\xi }$ are the propagation constants of the modes. ω is the angular frequency. The integration in (2) is performed using the variable
(3)$$\xi =\sqrt{n_{3}^{2}k_{0}^{2}-\beta_{\xi }^{2}},$$
where $k_{0}=\omega /c$ is the wavenumber and c is the speed of light in a vacuum. From Maxwell’s equations, it is possible to derive the following coupled mode equations for the core and radiation mode amplitudes:(4)$$\begin{array}{c} \frac{dc_{\mathrm{co}}}{dz}=-i\int g_{c\xi }\exp\left[i\varphi_{c\xi }z\right]c_{\xi }\left(z\right)d\xi ,\\ \frac{dc_{\xi }}{dz}=-ig_{\xi c}\exp\left[-i\varphi_{c\xi }z\right]c_{\mathrm{co}}\left(z\right)d\xi , \end{array}$$
where
(5)$$\varphi_{c\xi }=\beta_{\mathrm{co}}-\beta_{\xi }-\frac{2\pi }{\Lambda },$$
is the detuning parameter describing the offset from the resonance. The coupling coefficient is expressed through the overlapping integral of two interacting modes.
(6)$$g_{ij}=\frac{\omega \varepsilon_{0}}{4P_{i}}\int {\Delta n\left(r\right)E}_{i}\left(r\right)E_{j}\left(r\right)2\pi rdr,$$
where $P_{i}$ denotes the power carried by the i-th mode (the LP_01_ core mode or the ξ-th radiation mode), $\varepsilon_{0}$ is the dielectric permittivity of a vacuum, and $\Delta n\left(r\right)$ is the RI modulation in the fiber with the LPFG. We assume that the RI modulation is a constant and different from zero only in the core.

We calculated the discrete propagation constant, the fields of the core mode of the fiber, and a continuous set of modes of the fiber as a function of the variable ξ. Then, we employed the coupled mode Equation (4) to simulate the transmission of light through the LPFG. The fiber and the grating were assumed to have the same parameters as in the previous section.

The results of simulation are presented in Figure 2. The RI of the ambient medium changed from 1.45 to 1.61 with an increasing step. We started from an RI value that was a little higher than the RI of silica. The full transmission spectrum in the range 1300–1650 nm exhibits four resonances at about the same wavelengths as in Figure 1. Further, we considered the two resonances (dips 3 and 4) in the range of 1400–1600 nm (Figure 2a), since these were more sensitive to RI changes in the external medium. As expected, the wavelength of the transmission dips did not change significantly. The amplitude of the dips increased with the RI. When the RI was high, there was good reflection at the cladding outer interface due to the Fresnel reflection, and the resonances were deep. When the RI was close to the RI of silica, the reflection vanished, which resulted in the disappearance of the dips.

In Figure 2b, the dependence of transmission in the centers of the resonances is presented as a function of the RI of the surrounding medium. The transmission drops with RI, and it lowers faster for Dip 4 than for Dip 3. There is no loss at the point where the RI is equal to 1.444.

## 5. Measurement of RI Sensitivity

In order to calibrate our LPFG for RI sensing, we measured its spectra when it was submerged in liquids with higher RIs than silica. We used a set of liquids with different RIs from Cargille (Cedar Grove, NJ, USA). The fiber was placed straight in a groove under low tension. Then, the groove was filled with a liquid and a spectrum was measured. After each measurement, we wiped the groove and the fiber using alcohol. Before the experiment with liquids, we measured the spectrum of the source and the spectrum of the LPFG in air. To obtain the transmission, we subtracted the LPFG spectrum from the source spectrum. The results are shown in Figure 3. 

The spectrum of the LPFG in air (Figure 3a, dashed curve) has two dips at 1443 nm and 1573 nm with transmissions going down to T=0.04 and 0.0, respectively. These dips have profound sidelobes, as one can expect for a standard LPFG. In our case, the sidelobes on the higher-wavelength side of both dips are larger, probably due to some inhomogeneity of the grating. 

We started the experiment by immersing the fiber with the LPFG in a liquid with an RI of 1.45 at 1550 nm, which is a little bit higher than the RI of silica. The spectrum changed a lot: the amplitudes of dip 3 and dip 4 decreased more than twice so that the transmission in the minima became 0.33 and 0.40, respectively, the dips broadened, the sidelobes were smoothed out, and the loss value at wavelengths between the resonance notches (insertion loss) increased significantly. The center wavelengths of the third and fourth dips shifted to longer wavelengths by 0.82 and 1.14 nm, respectively. Then, we increased the RI of the liquid in steps of approximately 0.02 from 1.45 to 1.608 at 1550 nm. Each time the dip amplitudes grew and the insertion loss decreased, the spectrum became more similar to the spectrum of the LPFG in air. Unlike the amplitudes, the resonance wavelengths were almost unchanged and remained closer to the resonance wavelengths for 1.45. We may also note that no sidelobes appeared.

Figure 3b shows the dependence of the minima of dips 3 and 4 on the RI of the surrounding medium. We can see that the dips go down with increasing RI, with the steepest descent at RIs approaching 1.444. For higher RIs, the curve becomes close to a linear relationship with gradients of −0.449 and −0.927 RIU^−1^ at RIs around 1.54 for dips 3 and 4, respectively. The theoretical spectra of the LPFGs (Figure 2a) and the dependencies of the amplitudes on RI (Figure 2b) agree with the experimental results. There is some discrepancy for dip 3, which has a flatter dependence for higher RIs and a steeper one for lower RIs compared to the theoretical curve. The reason for this discrepancy can be due to the deviation of the RI modulation profile in the LPFG from a rectangular shape, which we assumed in our simulation.

The data can be well fitted by a curve represented as a sum of hyperbolic and linear functions with four coefficients, which are found to obtain the best fit. The numerical values of the coefficients are shown in an inserted table (Figure 3b). The same function, but with different parameters, can be used to fit the simulated data in Figure 2b. These fitting curves and the coefficients can be used for finding the RI of the surrounding medium from the amplitudes of the LPFG dips.

## 6. Monitoring of Degree of Curing

An experimental setup was developed to monitor the epoxy curing process. The setup consisted of a fiber optic broadband light source (EG&G Model OP507 1550, Boston, MA, USA), an optical spectrum analyzer (Anritsu MS9740B, Atsugi, Japan), a long-period fiber grating, K-type thermocouples and a Thermocouple Data Logger system (TC-08 from Pico Technology, Eaton Socon, UK) as shown in Figure 4. EL2 Epoxy Laminating Resin and AT30 Slow Epoxy Hardener from EasyComposites (Rijen, The Netherlands) were mixed. Light for the broadband source was transmitted through the LPFG connected by standard fibers to the optical spectrum analyzer.

The LPFG was placed in the center of the mold (80 × 26 × 3 mm) in a straight position, and both sides of the mold passing the fiber were sealed using tacky tape to avoid any epoxy leak from the mold. A thermocouple was placed close to the center of the LPFG to monitor the epoxy temperature variation during the curing process. The mold was mounted on an aluminum plate. The plate was placed on a heater with controlled temperature. Data from the thermocouple were acquired using the Thermocouple Data Logger system every second. The spectrum of the LPFG was acquired by the OSA every 2 min. The spectrum range, the resolution, and the number of sampling points were 1400–1600 nm, 1 nm, and 1001, respectively.

First, we measured the spectra of the LPFG in the range 1400–1600 nm without any resin, in air. It is shown in Figure 5 by the black solid curve. Two dips (dips 3 and 4) fall into the observed range and have the standard form. Then we filled the cuvette with a mixture of resin and hardener. The spectrum moved upwards (red solid curve), changing as we described above, which indicates that the RI of the mixture is higher than the RI of the silica cladding. 

The whole process of resin curing consisted of two stages: the first stage lasted 28 h at room temperature, and the second stage lasted 6 h at 60 °C, as recommended by the manufacture of the EL2 Epoxy Laminating Resin. The change in temperature during the curing is shown in Figure 6. In the first stage, there was some variation in temperature related to the variation in air temperature in the laboratory. This variation was too small to have any effect on the curing process. The curing is an exothermic reaction and is usually accompanied by the heating of the sample. In our case, a small volume of resin was used and the mold was intentionally mounted on an aluminum plate with good heat dissipation. Therefore, no elevation of temperature was observed in the first stage. After 28 h of curing, we raised the temperature of the mold to 60 °C. There was some overshooting at the beginning of the second stage, which is exhibited by the spike at the edge of the plateau in Figure 6. The heating stage was followed by cooling to room temperature.

Figure 5 demonstrates the evolution of the LPFG spectra during the whole experiment. The spectrum of the grating in air is shown by the solid curve (1), which has strong oscillations between the main resonances. Curve (2) corresponds to the case when the grating is in the resin at the beginning of room-temperature curing. We can see that most oscillations disappear and the amplitudes of the main resonances decrease significantly accompanied by a notable long-wavelength shift. The spectrum after 28 h of curing at room temperature is shown by the dashed curve (3). We can see an increase in the dip amplitude induced by increasing the RI of the mixture. The final spectrum after 6 h of curing at 60 °C is shown by the blue dashed curve (4). There is some additional increase in the dip amplitude together with a slight wavelength shift.

During the curing process, the wavelengths and amplitudes of dips 3 and 4 continuously changed with time. Figure 7 shows these dependencies for the two stages. It can be seen that the wavelength shifts (Figure 7a) are related to the temperature change in the resin. The amplitudes of the dips (Figure 7b) go down during the first stage of curing at room temperature. This decrease is related to the increasing RI of the resin as a result of curing. The curing rate quickly grows just after mixing, and reaches its maximum between 200 and 300 min later. Then, it slows down significantly after 900 min. 

In the second curing stage, the heating shifts the dips sharply upward, which is probably related to the decreasing RI, which is induced by the thermal expansion of the resin. Then, the dips go down (with temperature being constant), which demonstrates that high temperatures promote fast curing. The curing slows down after 100 min and is almost finished 150 min after the start of the heating cycle. The final cooling also reduces the dip amplitudes. 

All the previous curing stages were performed with the resin attached to the aluminum plate, which was thick enough to withstand the strain appearing in the resin. Therefore, the resin remained in a strained state when attached to the plate. In order to see the effect of this strain on the LPFG, we also measured the spectra of the LPFG after the detachment of the resin from the plate. Before the experiment, the plate was covered with a release agent, and the resin slab was detached by inserting a wedge between the slab and the plate. The resulting spectrum is demonstrated in Figure 8. It was compared to the spectra before the detachment and after an additional 24 h of curing. The detachment resulted in a shift in resonance dips to longer wavelengths and lower transmissions, the latter being related to an increase in the RI of the resin. The wavelength shift was probably produced by compressive longitudinal strain in the LPFG. Prolonged curing during the following day added to the amplitude of dip 4, while keeping dip 3 almost unchanged. This may be attributed to the minor additional curing of the resin.

## 7. Discussion

In Section 5, we showed that the amplitude of the dips of the LPFG depends on the RI of the surrounding medium. We fitted this dependence as a sum of linear and hyperbolic functions (shown in Figure 3b):(7)$$T=A+\frac{10^{-3}B}{n-C}-Dn,$$
where T is the transmission coefficient in the dip minimum. If we measure this transmission coefficient, then we can solve the inverse problem and find the RI of the resin by solving the fitting Equation (7), which can be represented in the form of a quadratic equation in terms of n:(8)$$Dn^{2}-\left(A+CD-T\right)n+AC-CT{-10}^{-3}B=0,$$

The solution to this equation is expressed by the following formula:(9)$$n=\frac{\left(A+CD-T\right)+\sqrt{{\left(A+CD-T\right)}^{2}-4D\left(AC-CT-10^{-3}B\right)}}{2D},$$

The coefficients A,B,C,D are given in Figure 3b for dips 3 and 4. The results of such a calculation are demonstrated in Figure 9. We also plotted an average of the two curves, since the difference between the RIs at the two wavelengths of dips 3 and 4 (1443 and 1573 nm) should be negligible. 

Assuming that the RI of the resin is proportional to its degree of cure and that the degree of cure at the end of the first curing stage is 75%, we can calculate the degree of cure, which is shown by the same curve with the scale on the right side of the plot. The isothermal degree of cure at 23 °C is derived through differential scanning calorimetry (DSC) measurements following the method proscribed in [33], with the total energy released at 23 °C over 48 h being taken as a complete cure [34]. Based on these data, the degree of cure at the end of the first curing stage (800 min) was found to be 75%. It can be seen that the curve has a short flat start followed by an increasing gradient, which reaches its maximum around 200 min after resin mixing. The curing slows at around 600 min and has virtually stopped at 850 min. From the difference between the curves for dips 3 and 4, we can estimate the accuracy of our measurements of the degree of cure as ±4%. 

The optical constants of fiber and resin material are wavelength-dependent. Therefore, we should take these dispersion properties into account when we simulate the spectra of LPFGs in a broad range of wavelengths. Some of them (such as thermo-optic and strain-optic) may not be accurately measured. In our case, we have taken into account the dispersion of the silica cladding of the fiber, doped silica core, resin, and immersion liquids. Considering dips 3 and 4, which are separated by 130 nm, the difference between the RIs of the resin at the two wavelengths is below $0.7\times 10^{-4}$ and the wavelength dependence can be neglected.

The second stage of curing is carried out at elevated temperatures. This makes the interpretation of LPFG spectra more complicated. In addition to the curing-induced RI change, the RI can be modified by thermo-optic and strain-optic effects, both in the LPFG itself and in the surrounding resin. As a result, the dip amplitude or wavelength in the LPFG spectrum may change.

Two effects may contribute to the amplitude change during cooling (Figure 7b): the first, the thermo-optic effect in the resin; the second, the increased strain of the resin and the resulting strain-optic effect. Two other main effects could change the resonance wavelengths but not the LPFG’s amplitudes: the thermo-optic effect in the LPFG and the transverse strain applied to the grating by the shrinking resin. Eventually, after the heat treatment, the dips have almost the same amplitudes as before the treatment. We may assume that the RI change produced by resin curing is compensated somewhat by the strain-optic effect. Compared to the initial amplitudes in the liquid mixture, the final amplitudes are significantly lower. 

Further study is required to thoroughly decouple the effects of the RI, temperature, and strain involved in the formation of LPFG spectra during the curing process of epoxy resins. In the process of resin curing, these three parameters change at the same time, which makes it difficult to extract each parameter separately. When the fiber with an LPFG is heated, the refractive index is changed due to the thermo-optic effect and the fiber is thermally expanded. When the fiber with an LPFG is strained, the refractive index is changed due to the strain-optic effect, the fiber is mechanically elongated, and the period of the grating increases. These changes in RI result in modifications of propagation constants of modes and shifts of resonance wavelengths. When the fiber with an LPFG is immersed in a medium with an RI different from the RI of air, the propagation constant of the cladding mode is increased due to the evanescent field of the mode propagating in the vicinity of the fiber cladding. The resonance wavelength shift depends on the fiber structure and the order of the cladding mode. All these effects should be taken into account to solve the decoupling problem. This would require measuring the response of the LPFG to different parameters separately and in different combinations. This is the subject of further research.

An additional problem is that the strain may be anisotropic, i.e., it can be different along different axes of the sample and the sensing fiber. This depends on the geometry of the sample, the geometry of the mold, and the infusion method. The anisotropic strain produces an anisotropic change in RI both in the fiber and the resin, which can contribute to changes in the optical spectra of the sensing LPFGs. 

## 8. Conclusions

We have demonstrated that LPFGs can be effectively used to monitor the change in the degree of curing of epoxy resins by observing the increasing refractive index (RI). By calculating the radiation cladding modes of the fiber and employing coupled mode theory, we simulated the modification of an LPFG transmission spectra as a function of the external RI, in the case when it is higher than the RI of the silica fiber cladding. We obtained the dependence of the amplitudes of the resonances of the long-period fiber gratings on the RI of the resin. We experimentally measured the sensitivity of amplitudes of the LPFG dips to the external RI. The theoretical spectra of the LPFGs and the dependencies of the amplitudes on the RI agree with the experimental results. For the range of RIs, corresponding to the RI of epoxy resins, we obtained a value of up to −0.927 RIU^−1^ for LPFG dips at wavelengths around 1550 nm. The degree of cure was calculated from the measured RI. The accuracy of the monitoring of the degree of cure was about 4% during isothermal curing. At the same time, the long-period fiber gratings were sensitive to temperature variations and internal strains induced during curing. These factors may affect the measurements of curing degree and should also be addressed. We also showed that high-temperature curing produces changes in the amplitudes and wavelengths of dips in the LPFG spectra, which were related to the thermo-optic and strain-optic effects in the resin.

## Figures and Tables

**Figure 1 sensors-24-03397-f001:**
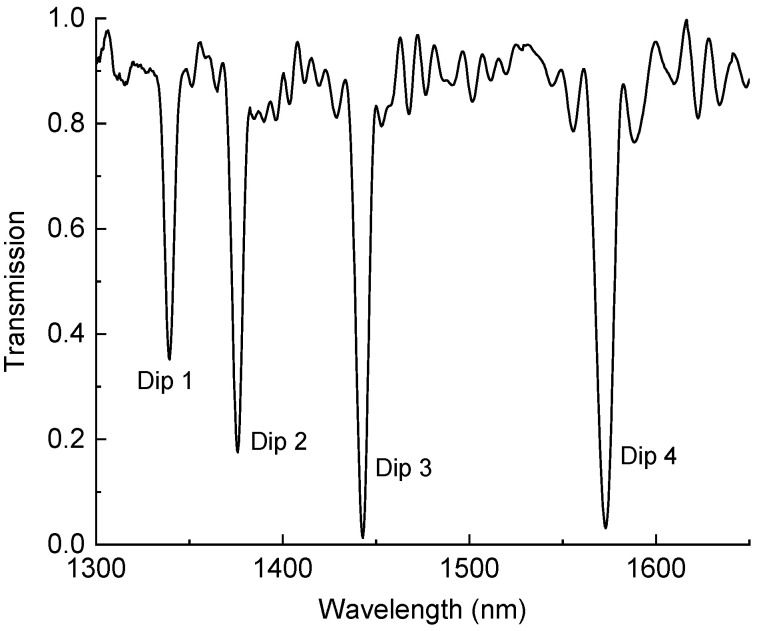
Transmission spectrum of LPFG with four cladding mode resonances.

**Figure 2 sensors-24-03397-f002:**
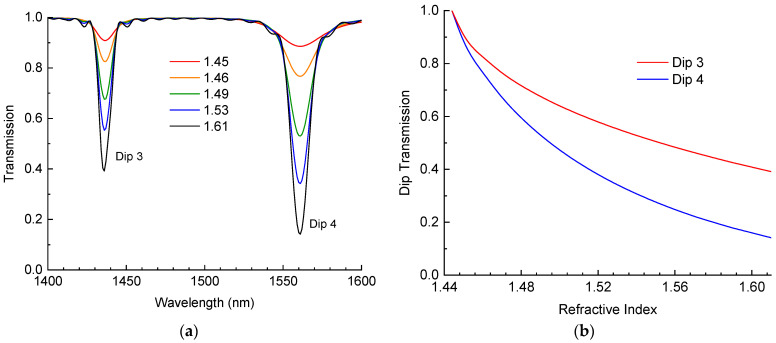
(**a**) Simulated spectra of two dips of LPFGs and (**b**) change in amplitude of the two dips vs. RI of surrounding medium.

**Figure 3 sensors-24-03397-f003:**
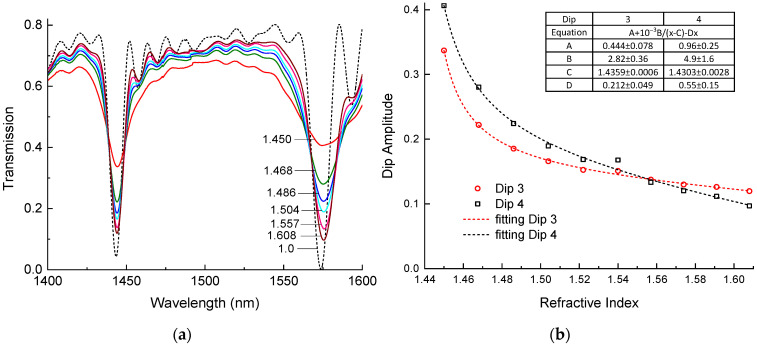
(**a**) Spectra of two dips of LPFGs and (**b**) change in amplitude of the two dips vs RI of surrounding liquid (measured at 1550 nm).

**Figure 4 sensors-24-03397-f004:**
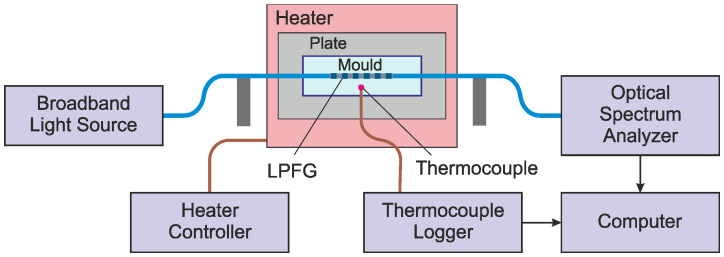
A scheme of the experimental setup used to monitor the epoxy curing process.

**Figure 5 sensors-24-03397-f005:**
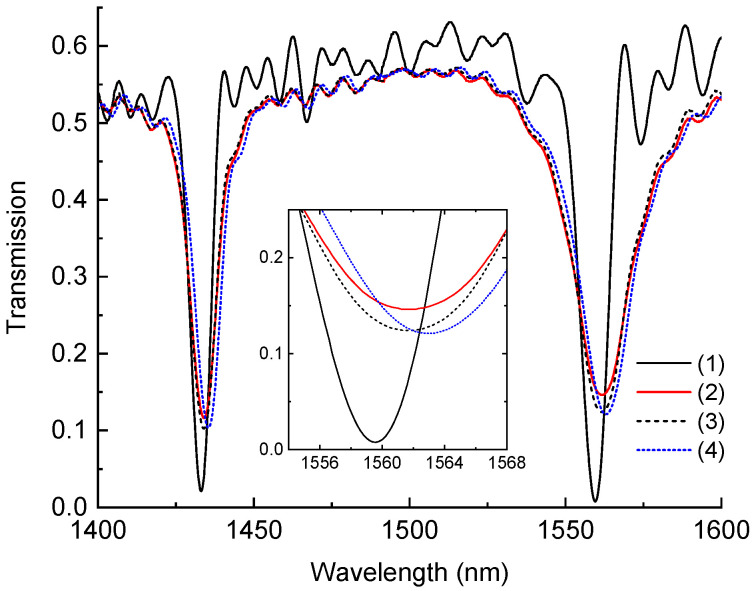
Spectra of the grating in air (1), in resin at the beginning of room-temperature curing (2), at the end of room-temperature curing (3), and at the end of high-temperature curing (4). Close view of the second dip is shown in the inset.

**Figure 6 sensors-24-03397-f006:**
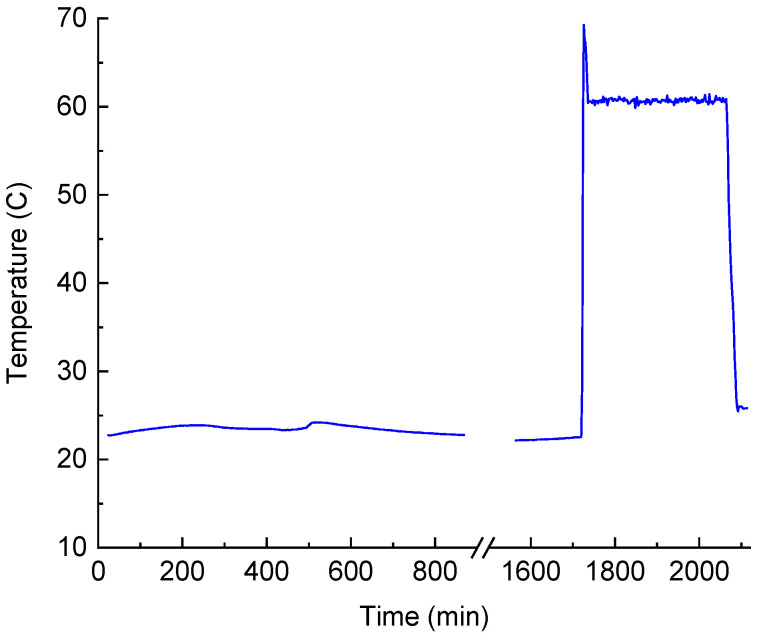
Temperature of the resin in the process of curing.

**Figure 7 sensors-24-03397-f007:**
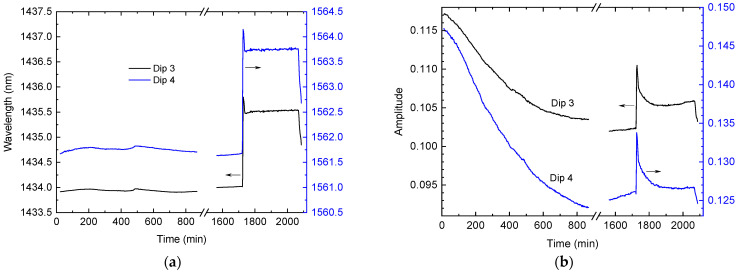
Dependences of (**a**) wavelengths and (**b**) amplitudes of dips 3 and 4 on time during the curing process. Note that the left-hand scale corresponds to the black line (dip 3) while the right-hand scale corresponds to the blue line (dip 4).

**Figure 8 sensors-24-03397-f008:**
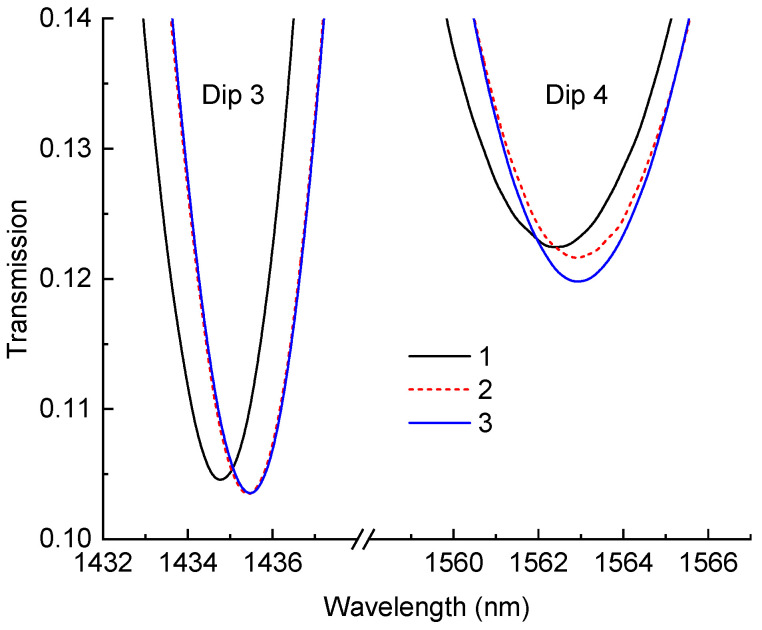
Spectra of LPFG in proximity of the dips after 35 h of curing (curve 1), before detachment, after 35 h of curing (curve 2), after detachment, and after 59 h of curing (curve 3).

**Figure 9 sensors-24-03397-f009:**
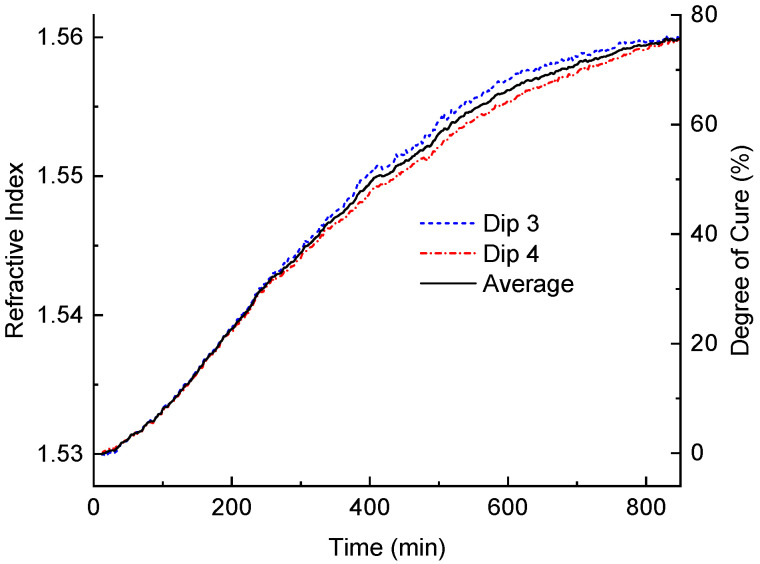
Variation in RI of the epoxy resin during room-temperature curing calculated from dip 3 (dashed curve), dip 4 (dash dot curve), and averaged (solid curve).

## Data Availability

The original contributions presented in the study are included in the article, further inquiries can be directed to the corresponding author.

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
