# Peer review of "Monitoring of Curing Process of Epoxy Resin by Long-Period Fiber Gratings"

_sensors, 2024, doi:10.3390/s24113397_

Round 1
Reviewer 1 Report
Comments and Suggestions for Authors
Reviewer Report
In this manuscript, the authors investigated the curing process of an epoxy resin by using LPFG. The refractive index of the epoxy resin increases during curing process and can be measured to determine the degree of curing. The Authors employed LPFG that are sensitive to the refractive index of an external medium for measurement of refractive index changes in the resin. They observed that the resonances of long-period fiber gratings increase their depth with increased refractive index of the resin, which is then described by simulations taking the coupling to radiation modes into account. The degree of cure can be estimated from the depth of the grating resonances using a phenomenological model. The present manuscript is suitable for publication in Sensors, subject to the following revision points:
1) In the introduction, authors should mention some more published works on the influence of radiation and bending on transmission in optical fibers, which is important for their potential sensing properties, such as:
- Measurements of growth and decay of radiation induced attenuation during the irradiation and recovery of plastic optical fibres, Optics and Laser Technology, Vol. 47, 2013, pp. 148-151.
- Theoretical investigation of bending loss in step-index plastic optical fibers, Optics Communications, Vol. 475, 2020, 126200 (4pp).
2) The origin of equation (8) should be provided.
3) Authors write: “Further study is required to decouple thoroughly the effects of RI, temperature, and strain involved in the formation of LPFG spectra during the curing process of epoxy resins.” Can the Authors anticipate at least how difficult would be to achieve this important task?
4) Authors should discuss the choice of the wavelength range investigated in this work.
5) English of the manuscript should be improved.
Comments on the Quality of English Language
Minor English corrections required
Reviewer 2 Report
Comments and Suggestions for Authors
This work focuses on the theme of in-situ monitoring of the curing process in the composite industry. The authors use integrated fiber optic sensors in this manuscript. The manuscript is well-written and presents research on refractive index (RI) measurements depending on various areas and temperature conditions. There are some issues to address before publishing.
1) There are only a few references from this decade. The authors should update their introductions by using up-to-date references.
2) In line 45, the authors mention that “These methods are mostly difficult to apply for real-time/in-situ measurements.” However, this is incorrect, as dielectric analysis is a real-time and in-situ method. The main advantage of fiber optic sensors is that they can be used as a damage control sensor in cured PCM.
3) As a result, the authors present a correlation between RI and cure degree (Figure 9). Why did the authors choose a cure degree of 75%? How was it determined? Why did they think that the actual cure degree would correlate with the average RI? This viewpoint should be confirmed by using over methods such as DEA, DMA, and TMA, or chemical methods, for example.
Round 2
Reviewer 1 Report
Comments and Suggestions for Authors
Accept
Reviewer 2 Report
Comments and Suggestions for Authors
The article has become more readable and may be published in its current form.